# A modified transfibular technique of ankle arthrodesis using partial fibular resection and onlay bone graft

Dong Yeon Lee[1☯‡]*, Min Gyu Kyung[1☯‡], Yun Jae Cho[2], Seongjae Hwang[1], Ho Won Kang[1], Dong-Oh Lee[1]

1 Department of Orthopedic Surgery, Seoul National University Hospital, Seoul, Republic of Korea,
2 Department of Orthopedic Surgery, Hanil General Hospital, Seoul, Republic of Korea

☯ These authors contributed equally to this work.
‡ These authors are co-first authors.
* leedy@snu.ac.kr

**Data Availability Statement:** All relevant data are within the manuscript and its Supporting Information files.

## Abstract

The transfibular approach is a common procedure for tibiotalar fusion. However, this technique has several concerns: inadequate stability to resist rotational and shearing forces, a fibula is suboptimal for bone grafting, and an onlay fibular graft that might prevent impacting and cause distraction. We present a modified transfibular technique using partial fibular resection and onlay bone graft, which may address these potential problems. This study aimed to evaluate whether the ankle joint is well fused with neutral alignment and functionally improved at the final follow-up. For this study, 27 consecutive patients (mean age, 68.5 years; range, 58–83) who underwent tibiotalar fusion with a follow-up period of >1 year were retrospectively included. A modified transfibular lateral approach was performed, in which the distal anterior half fibula was resected and fixed as an onlay graft to achieve fusion between the tibia, fibula, talus, and fibular onlay graft simultaneously. Radiographic outcomes were assessed using computed tomography at 4 months after operation and serial follow-up radiographs. Functional outcomes were assessed using the American Orthopedic Foot and Ankle Society hindfoot scale and Foot and Ankle Outcome Score. The mean follow-up period was 17.3 (range, 12–32) months. Four months after operation, complete union was achieved in 13 patients, near-complete union in 8 patients, and partial union in the remaining 6 patients. At the final follow-up, all the patients achieved complete union and maintained neutral ankle alignment. The functional outcome showed a significant increase between the preoperative and postoperative periods. One minor complication occurred, in which medial side ankle pain was relieved after screw removal. This modified technique is safe and effective, and has several merits, including saving the soft tissue of the anterior ankle, saving the course of the peroneal tendons by leaving the posterior half of the fibula, resected fibula serving as a good bone stock, and reducing the likelihood of valgus deformity after fibulectomy.

**Funding:** This study was supported by a grant (NRF-2017M3A9E2063104) from the Bio & Medical Technology Development Program of the National Research Foundation (NRF) funded by the Ministry of Science & ICT, Republic of Korea. The funders had no role in study design, data collection and analysis, decision to publish, or preparation of the manuscript.

**Competing interests:** The authors have declared that no competing interests exist.

## Introduction

For the treatment of patients with end-stage ankle osteoarthritis, ankle arthrodesis has been considered as a standard treatment for a long time [1]. Although total ankle arthroplasty has emerged as a promising option for patients with ankle osteoarthritis, ankle arthrodesis is still a viable option because it has merits of better pain relief [2] and a lower revision rate in spite of sacrificing the tibiotalar joint motion [1].

The fusion rate in ankle arthrodesis has been as high as 85% to 100% [3–6]. Numerous surgical procedures and various fixation methods have been introduced for tibiotalar fusion. The surgical approaches include transfibular, medial, anterior, and posterior [3, 7–9]. The fixation methods range from the use of multiple cannulated screws to that of a plate with screws and intramedullary nails, and sometimes, the Ilizarov method in complex cases [2, 9–11]. The transfibular approach has been one of the most widely used surgical procedures in tibiotalar fusion, with several advantages: it is technically simple, the joint is well visualized, and good reproducible outcomes have been reported [12, 13]. However, several concerns might arise with this technique, including inadequate stability to resist rotational and shear forces at the fusion site, the fibula being predominantly cortical bone and not suitable for bone grafting, and the possibility that fixation of the fibula to both the tibia and talus might prevent impacting and cause distraction [14, 15]. Furthermore, although the reported fusion rate was not substantially different among the surgical approaches in ankle arthrodesis [16, 17], in cases of nonunion or delayed union after fibular resection, valgus malalignment and peroneal tendon irritation can be problematic [16, 18].

In this study, we present a modified transfibular technique of ankle arthrodesis using partial fibular resection and onlay bone graft, which may address the possible issues described earlier. This study aimed to evaluate whether the ankle joint is well fused with neutral alignment and any adjacent joint is involved in possible arthritis. In addition, we investigated whether the patients showed functional improvements at the final follow-up as compared with their preoperative states. We hypothesized that by using our modified technique, we could expect excellent tibiotalar fusion 1 year after operation, with minimal complications and satisfactory functional outcomes.

## Materials and methods

### Study patients

This study was designed as a retrospective level IV case series. Twenty-seven consecutive patients who underwent tibiotalar fusion using our technique between May 2015 and December 2018 and followed-up for >1 year were included in this study. The modified transfibular technique of ankle arthrodesis described herein is considered a standard of care in our institution; therefore, no control group was included in the present study. Patients with septic arthritis, failed ankle arthroplasty, and neuropathic arthritis were excluded. All the enrolled patients visited the outpatient clinic with a chief complaint of persistent ankle pain, with end-stage tibiotalar joint arthritis on plain radiography. Before the surgical treatment, all the patients were treated conservatively using an ankle brace and medications for >6 months. If intolerable pain around the ankle and functional disability persisted, they were treated with ankle fusion using our technique. In the 27 patients, the diagnosis was posttraumatic arthritis in 18 patients, primary arthritis in 7, and rheumatoid arthritis in 2. The study subjects provided informed consent, and this study was approved by the Seoul National University Hospital Institutional Review Board (No. H-1806-151-953).

## Surgical technique

Under spinal anesthesia, the patient was placed in the lateral position, and skin preparation and draping were performed. The pneumatic tourniquet was inflated just before the start of the operation. The lateral malleolus was palpated, and an approximately 7- to 8-cm single longitudinal incision was made over the distal fibula. Careful dissection was performed to avoid injury to the sural and superficial peroneal nerves. The distal anterior half of the fibula approximately 6 cm from the tip was marked and resected with an oscillating saw (Fig 1). The resected fragment was kept separately for later steps. The ankle joint was visualized, and the osteophyte was removed. A lamina spreader was used to expose and distract the joint space (Fig 2). Fusion bed preparation was performed by removing denudated cartilage using a curette and burr. This preparation step was performed until a sign of subchondral bone bleeding was identified. After meticulous irrigation, microfracture using a hook osteotome was performed to keep the chopped cancellous bone between the tibia and talus interval.

Under intraoperative fluoroscopy, the alignment was checked, and the ankle position was set as neutral. Two temporary Kirschner wires were inserted to maintain the position, parallel from the talus to the tibia posteromedially. Then, another temporary Kirschner wire was inserted from the posterior side of the tibia to the talus anteriorly. A 6.5-mm cannulated screw with a washer, also known as a homerun screw, was inserted along the guide wire from the posterior to the anterior side, while two 6.0-mm headless compression screws (HCS) were inserted from the lateral to the medial side.

Prior to fixation of the previously resected anterior half of the fibula, the medial cortical bone was removed to expose the cancellous bone portion, and the upper and lower ends were removed to avoid reaching the level of the subtalar joint (Figs 3 and 4). The lateral side of the distal tibia and talus was decorticated, and the resected fibula was fixed as an onlay graft with three 2.7-mm cortical screws to obtain fusion between the tibia, fibula, talus, and fibular onlay

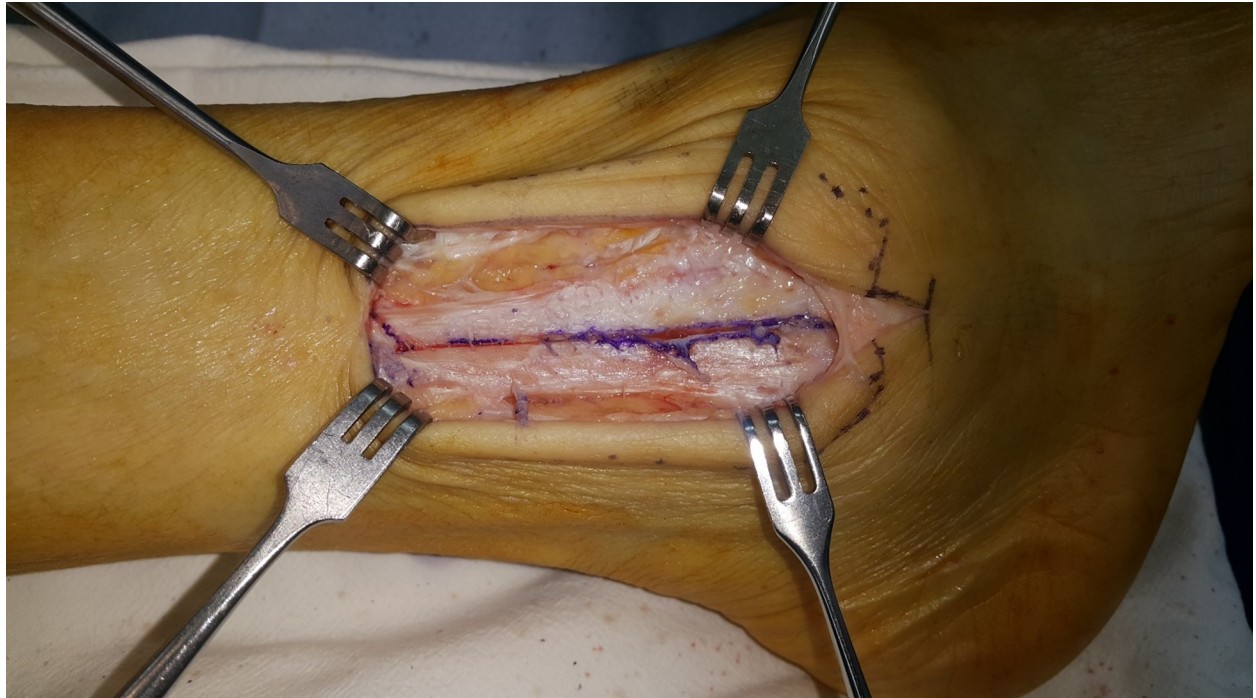

**Fig 1. The distal anterior half fibula osteotomized with a saw.** Note the gap at the proximal part showing the anterior half.

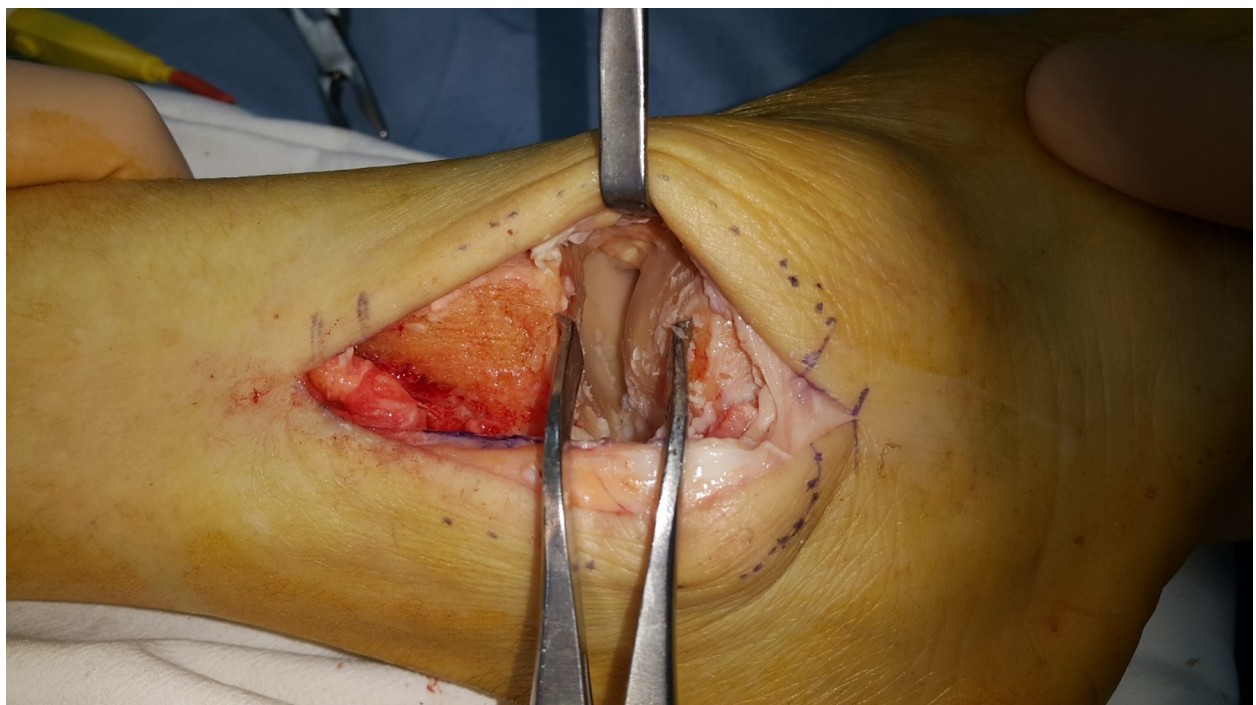

**Fig 2. Exposure of the tibiotalar joint space with a lamina spreader before preparation.**

graft simultaneously (4-in-1 procedure; Fig 5). The remaining chopped bone was placed between the spaces at the site of fusion, as an autologous bone graft. Fig 6 illustrates the concept of the overall procedures, and Fig 7 shows the immediate postoperative radiograph.

After irrigation and insertion of a drain, the subcutaneous tissue and skin were closed with Vicryl and nylon, respectively. Compressive dressing using plaster splints was performed to maintain stability.

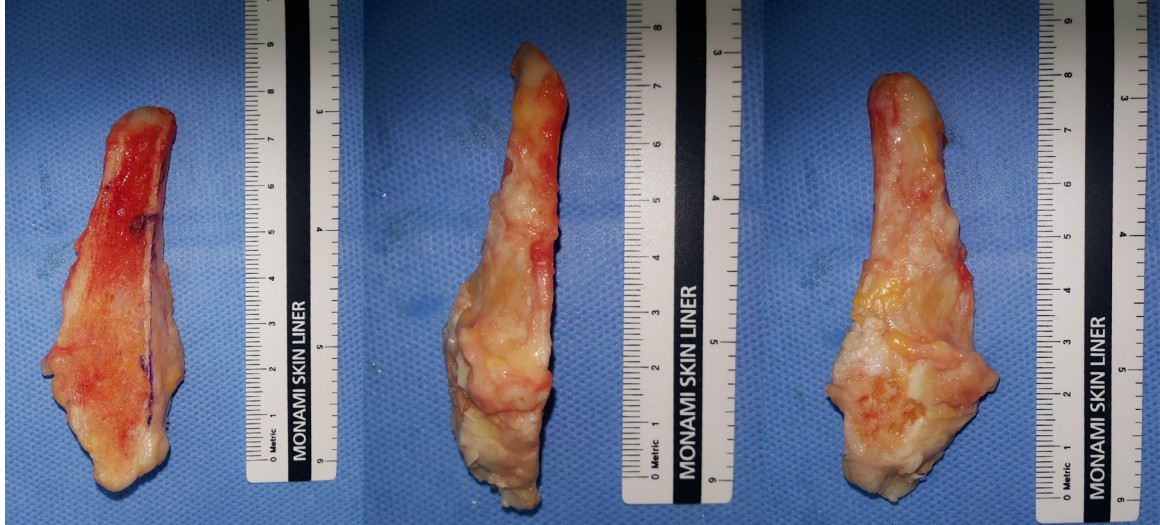

**Fig 3. Resected distal anterior half fibula.**

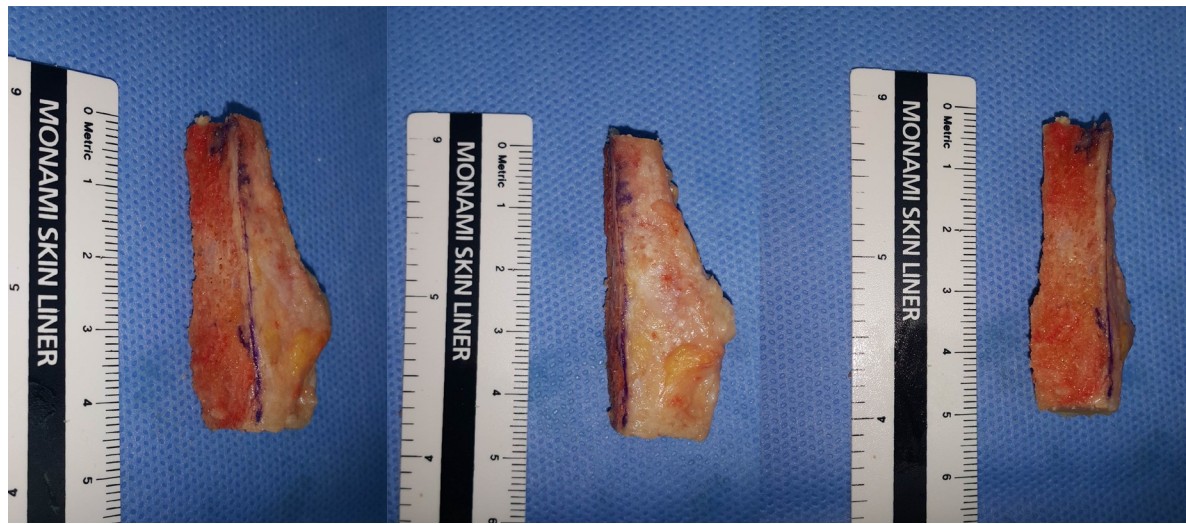

**Fig 4. The medial side of the resected fibula osteotomized to expose the cancellous bone portion.**

## Postoperative management

At postoperative 2 weeks, the wound was inspected, and the nylon suture was removed. A short leg cast was applied for 1 month postoperatively. During that period, the patients were allowed to weight-bear partially with crutches. Consecutively, the short leg cast was removed

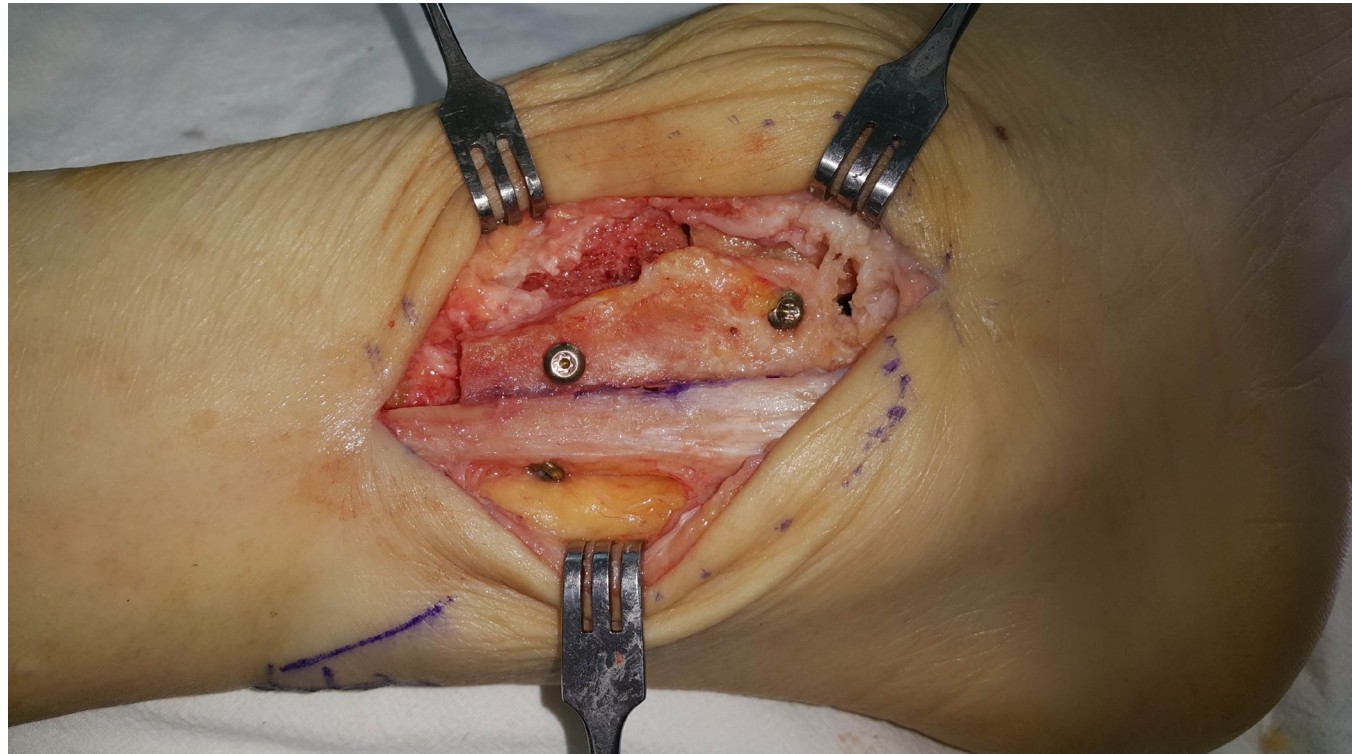

**Fig 5. After fixation of the partial fibular onlay graft with cortical screws.**

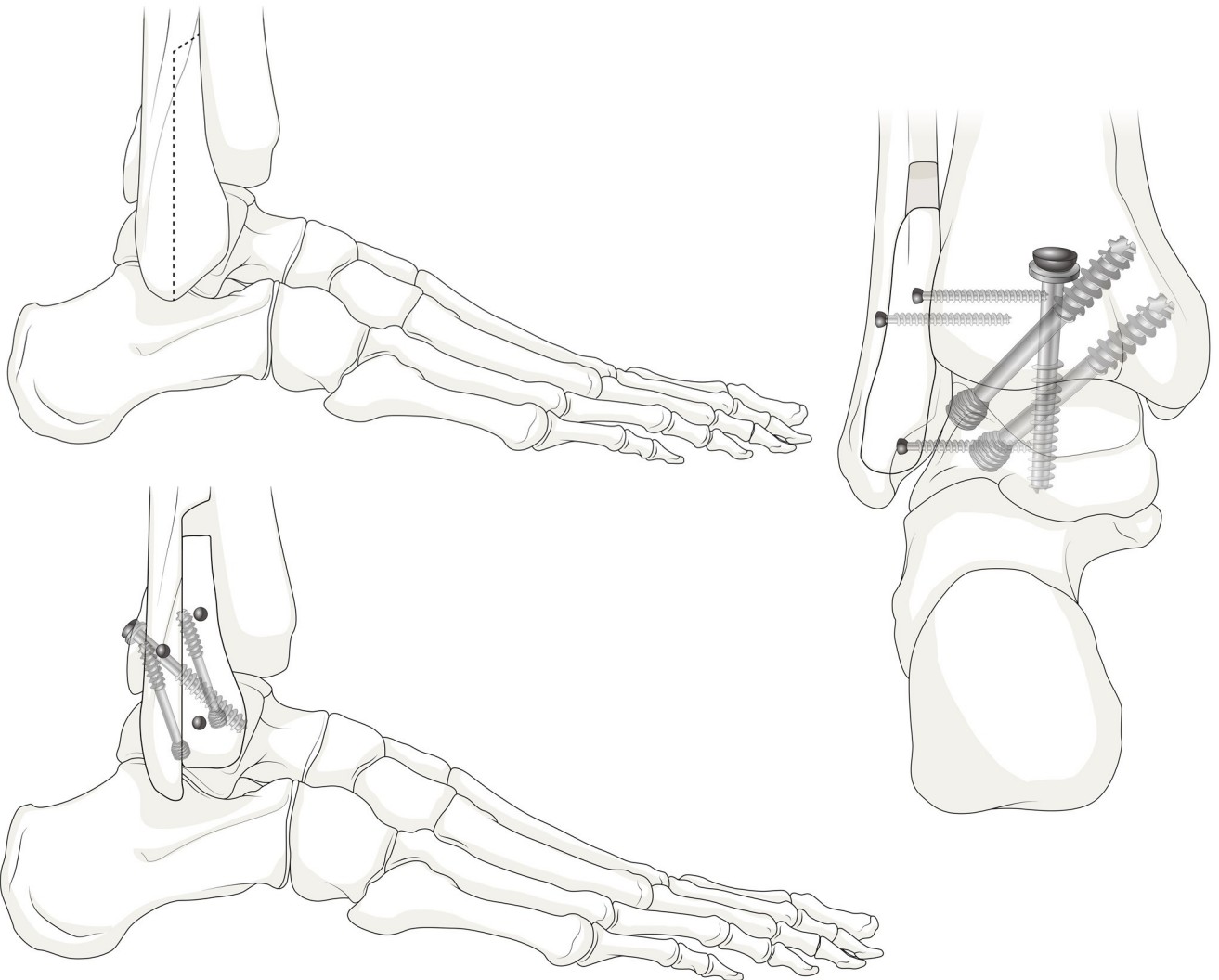

**Fig 6. Our modified surgical technique.**

and changed to a short leg yogips splint with an ankle brace for another month. The patients were also allowed to weigh-bear partially and encouraged to perform ankle range of motion (ROM) exercise. Then, the patients were instructed to wear an ankle brace and sustain activities of daily living with full weight-bearing until 6 months after operation.

Serial follow-up at the outpatient clinic was performed 2 and 4 weeks, and 2, 4, 6, and 12 months after operation. Plain ankle radiographs were taken at every visit. A postoperative ankle computed tomography (CT) scan was obtained once at 4 months after the operation.

## Radiographic measurement

Serial ankle plain radiographs and a CT scan, which was obtained at 4 months after operation, were assessed to check the alignment and bony union status. Bony union was confirmed radiographically by observing the presence of trabecular lines between the tibia and the talus at the point of contact, and the disappearance of the radiolucent line [19]. Partial union was defined as partial osseous bridging formation (<70%) in the tibiotalar joint but with a significant

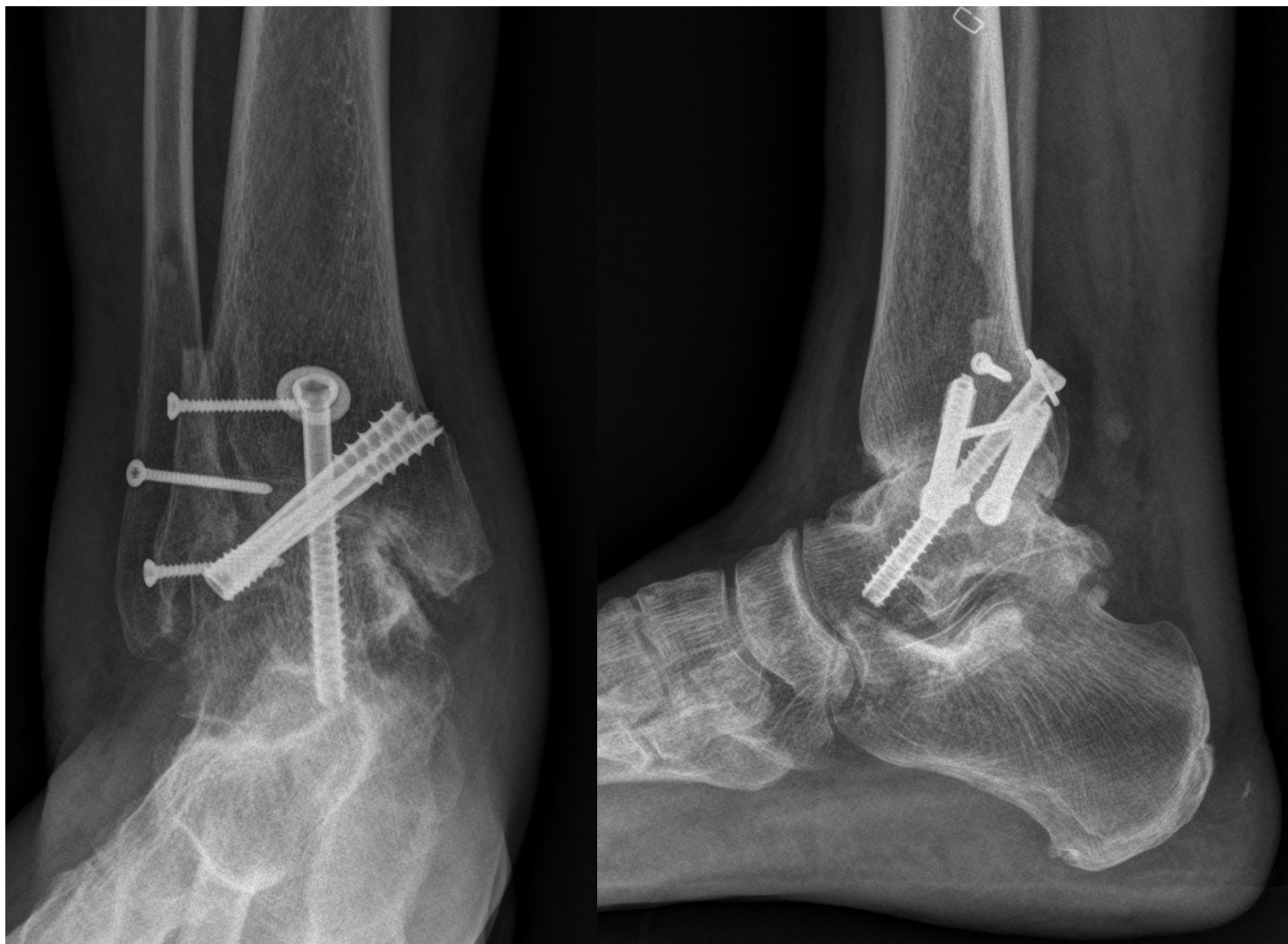

**Fig 7. Immediate postoperative ankle anteroposterior and lateral radiographs.**

radiolucent gap. Near-complete union was defined as a demonstrable osseous bridging in the sagittal and coronal views but with no complete absence of a radiolucent line.

The coronal tibiotalar angle was examined for postoperative valgus deformity. This angle was defined as the superomedial angle between the longitudinal axis of the tibia (created by connecting two points in the middle of the proximal and distal tibial shafts) and the axis of the talus (a line drawn through the talar shoulders) [20]. In addition, the adjacent talonavicular and subtalar joints were evaluated for subsequent arthritis. Adjacent joint arthritis was defined as the appearance of joint space narrowing or osteophyte formation on standing foot and ankle radiographs. The radiographic measurements were performed by two fellowship-trained orthopedic surgeons.

### Clinical outcome assessment

Functional outcome was evaluated using the American Orthopedic Foot and Ankle Society (AOFAS) hindfoot scale and Foot and Ankle Outcome Score (FAOS). FAOS was divided into several categories, including symptoms, pain, sports, activities of daily living (ADL), and

**Table 1. Patient demographic data.**

| Age, year | 68.5 (range, 58–83) |
|---|---|
| Sex, number | Male 14, Female 13 |
| Side, number | Left 12, Right 15 |
| Body Mass Index (BMI), kg/m$^2$ | 26.8 (range, 23.3–36.8) |

quality of life (QOL). These surveys were conducted preoperatively and postoperatively at the final follow-up.

## Statistical analyses

IBM SPSS Statistics 25 (New York, USA) was used for the statistical analysis. The Kolmogorov-Smirnov test was used to determine the normal distribution of data. A paired *t* test was performed to evaluate the difference between the preoperative and postoperative statuses. A P value of <0.05 was considered statistically significant.

## Results

The patients' demographic data are shown in Table 1. The mean follow-up period was 17.3 months (range, 12–32 months).

On the basis of the CT scan at 4 months after operation, complete union was achieved in 13 patients (Fig 8) and near-complete union in 8 patients. Partial union was observed in the remaining 6 patients.

At 6 months after operation, complete union was achieved in 20 patients and near-complete union in 6 patients. Partial union was found in 1 patient (Table 2). At 12 months after operation, complete union was achieved in 26 patients and near-complete union in 1 patient. At the final follow-up, complete union was achieved in all 27 patients (Fig 9).

With regard to the coronal alignment, all 27 patients had a neutral alignment at the final follow-up (Table 3).

Functional outcome data revealed a significant increase in score between the preoperative and postoperative periods in both the AOFAS hindfoot scale and overall FAOS (Table 4). Although no statistically significant difference was found with regard to the FAOS sports, an increasing trend was observed after the operation.

One minor complication occurred. The patient complained of ankle pain on the medial side during gait. After screw removal, the symptoms subsided. Otherwise, none of the cases had nonunion, wound infection, or metal failure requiring further operation. In addition, no newly developed adjacent joint arthritis such as talonavicular or subtalar joint arthritis, was found at the final follow-up.

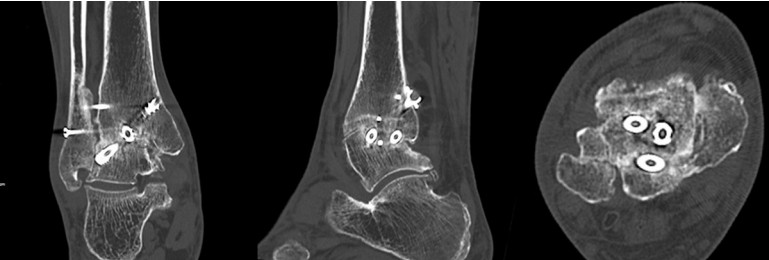

**Fig 8. Computed tomography images showing complete union 4 months after operation.**

**Table 2. Union rate assessed using plain radiographs at 6 and 12 months after operation and the final follow-up.**

|  | Postoperative 6 months | Postoperative 12 months | Final follow-up |
|---|---|---|---|
| Complete union | 20 | 26 | 27 |
| Near-complete union | 6 | 1 | 0 |
| Partial union | 1 | 0 | 0 |

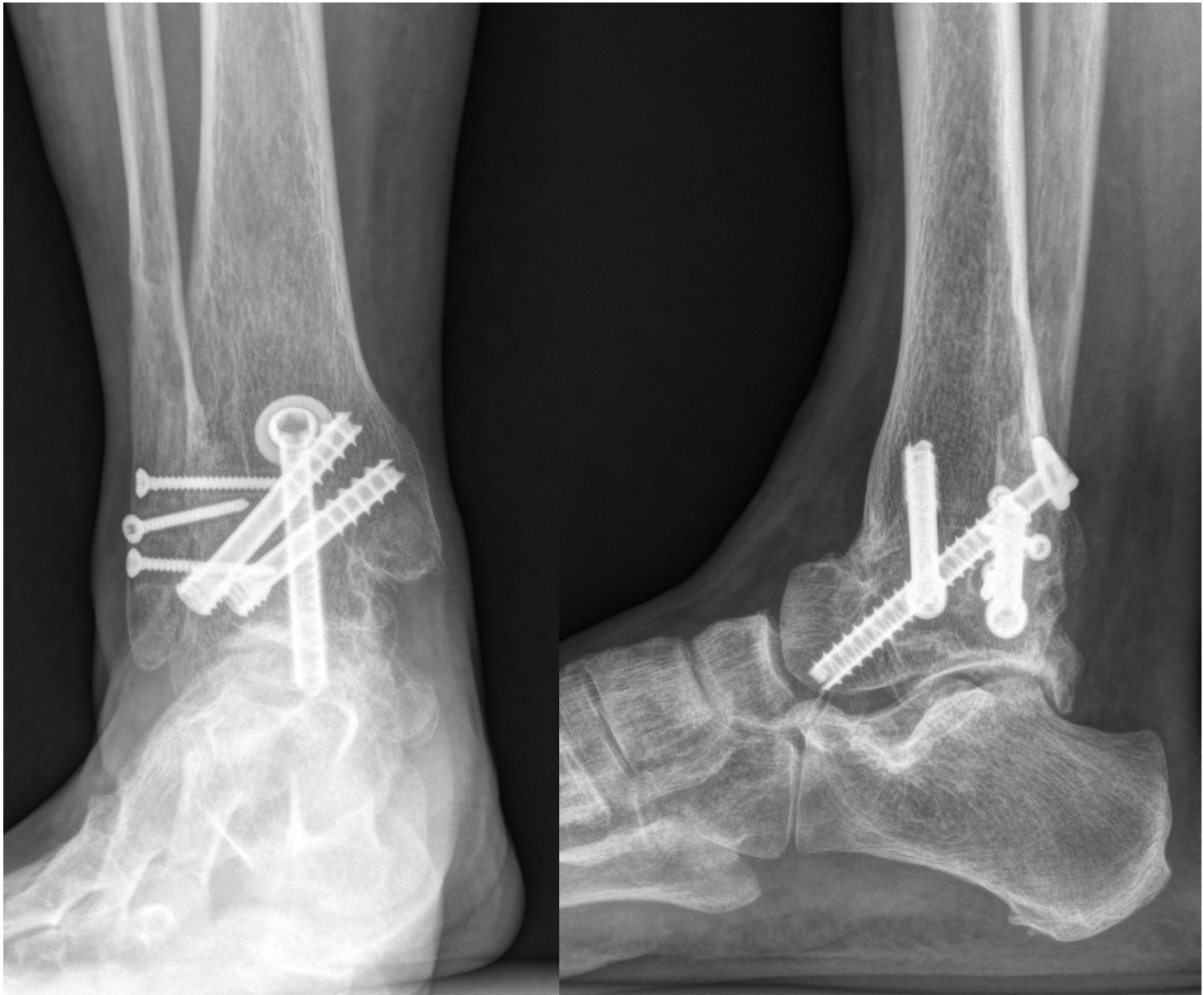

**Fig 9. Ankle anteroposterior and lateral radiographs at the final follow-up (13 months after operation).**

**Table 3. Coronal alignment before and after ankle arthrodesis.**

|  | Preoperative | Postoperative 6 months | Postoperative 12 months | Final follow-up |
|---|---|---|---|---|
| Coronal tibiotalar angle, degrees | 82.25 (57.33–105.25) | 88.94 (84.32–94.40) | 88.95 (84.50–90.30) | 88.94 (84.50–90.30) |

Data are presented as mean (range).

**Table 4. Functional outcomes in the preoperative period and postoperative final follow-up.**

|  | Preoperative | Final follow-up | P value |
| --- | --- | --- | --- |
| FAOS symptom | 61.96 (25–93) | 77.33 (50–100) | 0.012 |
| FAOS pain | 57.96 (19–89) | 85.83 (56–100) | <0.001 |
| FAOS ADL | 56.76 (19–87) | 88.50 (69–100) | <0.001 |
| FAOS sports | 19.42 (0–75) | 35.00 (0–75) | 0.067 |
| FAOS QOL | 24.46 (0–75) | 64.11 (13–100) | <0.001 |
| AOFAS hindfoot scale | 58.80 (22–89) | 79.83 (53–100) | <0.001 |

Data are presented as mean (range).

Abbreviations: FAOS, Foot and Ankle Outcome Score; ADL, Activities of Daily Living; QOL, Quality of Life; AOFAS, American Orthopedic Foot and Ankle Society.

## Discussion

Ankle arthrodesis using our modified fibular approach (4-in-1 union technique using the anterior half of the fibula) showed not only a promising fusion rate but also a good clinical outcome.

For a long time, ankle arthrodesis has been considered as a standard treatment for patients with end-stage ankle osteoarthritis, with the merits of good pain relief, stable plantigrade foot, and lower revision rate [1, 2]. Therefore, despite the improvements in implants and techniques in total ankle arthroplasty over the last decade, arthrodesis remains a good option. Although ankle arthrodesis has several merits as compared with total ankle arthroplasty, nonunion is one of the major complications in tibiotalar fusion [1].

Various surgical approaches have been used for performing ankle arthrodesis, including anterior, transfibular, and even arthroscopic approaches. Rowan and Davey reported high union rates and excellent clinical results of ankle fusion with an anterior plate [9]. Arthroscopic ankle arthrodesis has the advantage of less soft tissue dissection, and good clinical outcomes have been reported [5].

Ankle fusion by the transfibular approach is known to be first introduced by Horwitz [21]. Later, other authors modified the technique, and various techniques to achieve fusion were developed. Several studies have reported good outcomes using the transfibular approach and cannulated screws. Lee *et al.* reported a method of arthrodesis using screws with a single lateral incision [22, 23]. Kim *et al.* compared the anterior and transfibular lateral approaches, and reported that at the time of final follow-up, the transfibular approach group showed statistically more valgus angulation of the ankle joint [16].

In our study, ankle arthrodesis using a modified transfibular approach showed a good fusion rate and satisfactory clinical outcome. We think that this technique has several advantages to the anterior and conventional transfibular approaches using total fibulectomy.

First, the soft tissue of the anterior ankle is preserved for later operation, if needed. Recently, conversion procedures from fused ankle to total ankle arthroplasty via the anterior approach have been reported with considerable success rates [24, 25]. Although the necessity for conversion from fused ankle to total ankle arthroplasty remains controversial, saving the soft tissue of the anterior ankle might be beneficial.

Second, because the posterior half of the fibula is preserved in our technique, the course of the peroneal tendons is not disturbed. Although the tibiotalar joint is fused and assumed to have less motion, the eversion motion at the subtalar joint by peroneal tendons is possible. In all previous transfibular approaches, the entire distal fibula was removed [12, 22, 23]. This may cause irritation and possible peroneal tendon synovitis. Even if the resected whole fibula is

fixed again as an onlay graft, it is not anatomical and may cause irritation. Smith *et al.* reported that in ankle arthrodesis with a fibular-sparing technique, an intact fibula provides additional surface area for fusion, blocks valgus drift, guides proper rotation, and maintains the native groove and restraints for the peroneal tendons [18]. In addition, another author reported that the peroneal tendons might lose their biomechanical fulcrum around which they act during eversion of the hindfoot after resection of the distal fibula [26].

Third, not only the resected fibula serves as a good bone stock, but also the tibiotalar joint surrounded by the cancellous portion promises a good fusion environment. Previous studies have shown that adequate bone grafting is an essential component to achieve good fusion [27]. On the basis of our data, even in substantial bone defect cases of severe preoperative varus deformity, bone from the resected fibula was a good source of bone graft, and none of the patients needed a separate incision for bone graft. Considering that cancellous bone heals by fast membranous bone formation, which is important for bony union [28], our 4-in-1 union technique using the anterior half of the fibula provided a good fusion bed, and cancellous bone-to-cancellous bone contacts between the tibia, fibula, talus and the fibular onlay graft were achieved simultaneously. This possible advantage of the 4-in-1 union procedure was previously described in the treatment of congenital pseudoarthrosis of the tibia [29]. We believe that if the onlay fibular graft had blocked fusion and caused a distraction between the tibia and talus, the good fusion results shown in our study would not have been achieved.

Finally, our technique showed good stability and alignment at the final follow-up. Maintaining the posterior half of the fibula intact may reduce the likelihood of subsequent valgus deformity of the ankle joint, whereas conventional total fibulectomy cannot prevent valgus deformity in cases of tibiofibular nonunion or delayed union [16].

On the other hand, some possible concerns remain. First, although our data showed good union rate between the tibia and talar interface, union problems at other interfaces such as the tibiofibular and talofibular space may arise. Thus, further studies are needed. Second, metal failure may occur. Lee *et al.* also reported a case of instrument breakage [22]. In such situations, removing the broken screw and refixing the screw may be needed, or revision surgery with another plate is an option. With regard to the revision, no single useful instrument has been established. As in our technique, the remnant posterior half fibula may serve as a reliable support. The shearing force may be limited by the remaining posterior fibula. Thordarson *et al.* pointed out the importance of fibular strut grafts, which provide additional stability and resistance to rotational forces [27]. Third, even if our modified technique showed no newly developed talonavicular joint or subtalar joint arthritis, this may occur in long-term follow-up [30–32], and subsequent adjacent joint arthrodesis may be needed. Lastly, use of our technique could lead to takedown of ankle fusion and conversion to total ankle replacement. Greisberg *et al.* reported the poor result of conversion to total ankle replacement when the lateral malleolus was resected at the time of previous fusion [33]. Our technique of preserving the posterior fibula may reduce the risk of complications when conversion to total ankle replacement is needed.

Therefore, we suggest that this modified technique is not only for general indications requiring the conventional transfibular approach, where poor soft tissue quality at the anterior ankle may be applicable, but also for patients who want to maintain the lateral malleolar contour. However, the contraindications of this technique might include soft tissue defects in the lateral malleolar area or cases of severe distal fibular deformity, for which anterior half fibulectomy and onlay graft fixation may not be feasible.

This study has several limitations. This was a retrospective study with a relatively small number of patients included. In addition, the follow-up period of this study was short, and the different follow-up periods made the comparison of functional outcomes difficult. Mid-term to long-term outcomes are warranted to clarify the merits of this novel technique.

## Conclusions

Our modified transfibular ankle arthrodesis technique using distal anterior half fibulectomy and onlay bone graft proved to be a safe and effective treatment for end-stage ankle osteoarthritis.

## Supporting information

**S1 Table. A list of patients' demographic data, follow-up period, and diagnosis.**
(XLSX)

**S2 Table. A list of union state after operation.**
(XLSX)

**S3 Table. A list of patients' coronal tibiotalar angle before and after ankle arthrodesis.**
(XLSX)

## Acknowledgments

We thank Medical Art Studio (Sun Joo Kim and Woohyun Cho) for the medical illustrations of our concept.

## Author Contributions

**Conceptualization:** Dong Yeon Lee, Min Gyu Kyung.

**Data curation:** Dong Yeon Lee, Min Gyu Kyung, Yun Jae Cho, Seongjae Hwang, Ho Won Kang.

**Formal analysis:** Dong Yeon Lee, Min Gyu Kyung.

**Funding acquisition:** Dong Yeon Lee.

**Investigation:** Dong Yeon Lee, Min Gyu Kyung, Yun Jae Cho, Seongjae Hwang, Ho Won Kang, Dong-Oh Lee.

**Methodology:** Dong Yeon Lee, Min Gyu Kyung, Dong-Oh Lee.

**Project administration:** Dong Yeon Lee.

**Supervision:** Dong Yeon Lee.

**Validation:** Dong Yeon Lee, Min Gyu Kyung, Seongjae Hwang, Ho Won Kang, Dong-Oh Lee.

**Visualization:** Min Gyu Kyung.

**Writing – original draft:** Dong Yeon Lee, Min Gyu Kyung.

**Writing – review & editing:** Dong Yeon Lee, Min Gyu Kyung, Yun Jae Cho, Seongjae Hwang, Ho Won Kang, Dong-Oh Lee.

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
