## [Decision Letter · Decision Letter 0]

18 Aug 2020

PONE-D-20-16145

A modified transfibular technique of ankle arthrodesis using partial fibular resection and onlay bone graft

PLOS ONE

Dear Dr. Lee,

Thank you for submitting your manuscript to PLOS ONE. After careful consideration, we feel that it has merit but does not fully meet PLOS ONE’s publication criteria as it currently stands. Therefore, we invite you to submit a revised version of the manuscript that addresses the points raised during the review process.

As both reviewers point out, your study is interesting. Please, follow the reviewers suggestions to improve your manuscript. Seek advice from a native speaker. Add a new structure to the abstract including a clear question, explain the study design, and mention the indications for your technique.

We look forward to receiving your revised manuscript.

Kind regards,

Hans-Peter Simmen, M.D., Professor of Surgery

Academic Editor

PLOS ONE

Journal Requirements:

2. Thank you for including your ethics statement :

"Study subjects have given informed consent and this study was approved by our Institutional Review Board."

Reviewers' comments:

Reviewer's Responses to Questions

**Comments to the Author**

1. Is the manuscript technically sound, and do the data support the conclusions?

Reviewer #1: Yes

Reviewer #2: Yes

2. Has the statistical analysis been performed appropriately and rigorously? 

Reviewer #1: Yes

Reviewer #2: Yes

3. Have the authors made all data underlying the findings in their manuscript fully available?

Reviewer #1: Yes

Reviewer #2: Yes

4. Is the manuscript presented in an intelligible fashion and written in standard English?

Reviewer #1: Yes

Reviewer #2: Yes

5. Review Comments to the Author

Reviewer #1: This study reports a series of cases that were treated by a modified technique for ankle fusion. While the techniques seems to be an interesting, the manuscript needs major revisons.

I would also recommend to have the manuscript edited by a native speaker.

Specific comments:

Title: ok

Abstract:

23: what was the scientific question of this study?

25: please add information on the study design (retrospective/prospective?)

25: please add information on patients’ age.

30: please replace “x-rays” by “radiographs” throughout the manuscript

42, “reducing the likelihood of valgus deformity”: valgus deformity was not assessed in this study and there was no control group. hence, this conclusion cannot be made.

Introduction:

46: “Until 21th centuries…”: Please rephrase the whole sentence.

59: “the fibula is predominantly … and cause distraction”: these problems are not really addressed by the modified technique, as well.

65: what was the scientific question of this study? What was the authors’ hypothesis?

Methods:

71: please add information on the study design (retrospective/prospective?)

134: the authors state that follow-up visits were made at 2 and 4 weeks, 2, 4, 6, and 12 months but the manuscript presents only radiographic data at 4 months and final FU and functional data at final-FU. The final follow-up varies significantly (12 to 32 months). It would be important to report fusion rates at 6 and 12 months, as well. That would also give us an idea on the effect of different imaging (CT vs radiographs) for the assessment of fusion.

I would also highly recommend to add measurements on valgus deformity at preop, postop, 6 months and 12 months as the authors claim this to be a potential advantage of their technique.

150: why not at 12 months? It is very difficult to compare functional outcome at “final FU” with patients that are 12 months postop and 32 months postop in the same group.

155: how did the authors test for normal distribution? I would argue that parametric tests are preferable in a study with n=27.

Results:

160, “maintained neutral ankle alignment”: how was this determined?

165: how were “near complete union” and “partial union” defined?

185: how was “talonavicular joint or subtalar joint arthritis” defined/assessed?

Discussion

Generally well written except for quite a number of language errors that should be edited by a native speaker.

Very nice figures!

Reviewer #2: I would recommend to look through the manuscript regarding some grammatical errors. There are minor, but manageable.

Despite this, I would add the proper indications for this technique. Of course, contraindications and what could be expected from transfibular fusions (i.e. is there any chance for a conversion back into TAR?).

The technique, however, is of great interest. In our institution we have used the same technique for years due to same concerns as reported by the authors.

I have to congratulate the authors to report their results and detailed technique.

6. PLOS authors have the option to publish the peer review history of their article (what does this mean?). If published, this will include your full peer review and any attached files.

Reviewer #1: **Yes: **Georg Osterhoff

Reviewer #2: No

---

## [Author Response · Author response to Decision Letter 0]

2 Oct 2020

30-September-2020

Response to Editor-in-Chief

Dear Editor:

We carefully reviewed the excellent comments of two reviewers on the revision of our paper. We actively reflected the additional two reviewers' comments in our paper. The responses to the reviewers' comments are well explained in ‘Response to reviewers’. We think this additional comment on the revision of our paper has helped to improve the quality of our paper.

Best regards,

Dong Yeon Lee, M.D.

Response to Reviewers

Thanks for the good comments on the revision of our study.

We revised our manuscript to fully reflect the opinions of two reviewers. The opinions of the two reviewers helped to make our manuscript a higher level.

Thank you for your kind consideration.

Sincerely,

Dong Yeon Lee, M.D.

Review Comments to the Author

Reviewer #1: 

- This study reports a series of cases that were treated by a modified technique for ankle fusion. While the techniques seems to be an interesting, the manuscript needs major revisions. I would also recommend to have the manuscript edited by a native speaker.

-> Thank you for your comment. The revised manuscript was polished by a native speaker.

Specific comments:

Title: ok

-> Thank you for your comment.

Abstract:

- 23: what was the scientific question of this study?

-> Thank you for your comment. Authors have revised the manuscript according to your recommendation as follows:

“We present a modified transfibular technique using partial fibular resection and onlay bone graft, which may address these potential problems. This study aimed to evaluate whether the ankle joint is well fused with neutral alignment and functionally improved at the final follow-up.”

- 25: please add information on the study design (retrospective/prospective?)

-> Thank you for your comment. Authors have revised the abstract according to your recommendation as follows:

“For this study, 27 consecutive patients (mean age, 68.5 years; range, 58–83) who underwent tibiotalar fusion with a follow-up period of >1 year were retrospectively included.”

- 25: please add information on patients’ age.

-> Thank you for your comment. Authors have revised the manuscript as follows:

“For this study, 27 consecutive patients (mean age, 68.5 years; range, 58–83) who underwent tibiotalar fusion with a follow-up period of >1 year were retrospectively included.”

- 30: please replace “x-rays” by “radiographs” throughout the manuscript.

-> Thank you for your comment. Authors have replaced the term according to your recommendation throughout the manuscript.

- 42, “reducing the likelihood of valgus deformity”: valgus deformity was not assessed in this study and there was no control group. hence, this conclusion cannot be made.

-> Thank you for your comment. Authors have added the table showing the coronal alignment in Table 3.

-> Accordingly, previous Table 3 has been changed to Table 4.

Introduction:

- 46: “Until 21th centuries…”: Please rephrase the whole sentence.

-> Thank you for your comment. The sentence was revised according to your recommendation as follows:

“For the treatment of patients with end-stage ankle osteoarthritis, ankle arthrodesis has been considered as a standard treatment for a long time [1]”

- 59: “the fibula is predominantly … and cause distraction”: these problems are not really addressed by the modified technique, as well.

-> Thank you for your comment. Our modified technique’s advantage opposing to those concerns in line 59 is well described in Discussion. Authors have revised the manuscript according to your recommendation as follows:

“Third, not only the resected fibula serves as a good bone stock, but also the tibiotalar joint surrounded by the cancellous portion promises a good fusion environment. Previous studies have shown that adequate bone grafting is an essential component to achieve good fusion [27]. On the basis of our data, even in substantial bone defect cases of severe preoperative varus deformity, bone from the resected fibula was a good source of bone graft, and none of the patients needed a separate incision for bone graft. Considering that cancellous bone heals by fast membranous bone formation, which is important for bony union [28], our 4-in-1 union technique using the anterior half of the fibula provided a good fusion bed, and cancellous bone-to-cancellous bone contacts between the tibia, fibula, talus and the fibular onlay graft were achieved simultaneously. This possible advantage of the 4-in-1 union procedure was previously described in the treatment of congenital pseudoarthrosis of the tibia [29]. We believe that if the onlay fibular graft had blocked fusion and caused a distraction between the tibia and talus, the good fusion results shown in our study would not have been achieved.”

65: what was the scientific question of this study? What was the authors’ hypothesis?

-> Thank you for your comment. Authors have revised the manuscript according to your recommendation as follows:

“In this study, we present a modified transfibular technique of ankle arthrodesis using partial fibular resection and onlay bone graft, which may address the possible issues described earlier. This study aimed to evaluate whether the ankle joint is well fused with neutral alignment and any adjacent joint is involved in possible arthritis. In addition, we investigated whether the patients showed functional improvements at the final follow-up as compared with their preoperative states. We hypothesized that by using our modified technique, we could expect excellent tibiotalar fusion 1 year after operation, with minimal complications and satisfactory functional outcomes.”

Methods:

- 71: please add information on the study design (retrospective/prospective?)

-> Thank you for your comment. Authors have revised the manuscript according to your recommendation as follows:

 “This study was designed as a retrospective level IV case series.”

- 134: the authors state that follow-up visits were made at 2 and 4 weeks, 2, 4, 6, and 12 months but the manuscript presents only radiographic data at 4 months and final FU and functional data at final-FU. The final follow-up varies significantly (12 to 32 months). It would be important to report fusion rates at 6 and 12 months, as well. That would also give us an idea on the effect of different imaging (CT vs radiographs) for the assessment of fusion.

-> Thank you for your comment. Authors have revised Table 2 showing union rate at postoperative 6, 12 months and final follow-up. 

- I would also highly recommend to add measurements on valgus deformity at preop, postop, 6 months and 12 months as the authors claim this to be a potential advantage of their technique.

-> Thank you for your comment. Authors have added Table 3 showing the coronal alignment before and after ankle arthrodesis.

- 150: why not at 12 months? It is very difficult to compare functional outcome at “final FU” with patients that are 12 months postop and 32 months postop in the same group.

-> Thank you for your comment. Authors agree that it would have been a better study if functional outcome scores were collected at the same period, such as postoperative 12 months. However, the study was designed as a retrospective manner and functional score could not be gathered and compared at the same chronological time. But there were only few patients with longer follow-up period (5 patients over 24 months). Therefore, we included this concern in the limitation of this study.

 “In addition, the follow-up period of this study was short, and the different follow-up periods made the comparison of functional outcomes difficult.”

- 155: how did the authors test for normal distribution? I would argue that parametric tests are preferable in a study with n=27.

-> Thank you for your comment. As Kolmogorov-Smirnov test showed normal distribution of our data, we used paired t-test (parametric) instead of Wilcoxon signed rank test (non-parametric). Authors revised the manuscript including this information as follows:

 “The Kolmogorov-Smirnov test was used to determine the normal distribution of data.”

Results:

- 160, “maintained neutral ankle alignment”: how was this determined?

-> Thank you for your comment. Authors have added the information in the Materials and methods section as follows:

“The coronal tibiotalar angle was examined for postoperative valgus deformity. This angle was defined as the superomedial angle between the longitudinal axis of the tibia (created by connecting two points in the middle of the proximal and distal tibial shafts) and the axis of the talus (a line drawn through the talar shoulders) [20].”

-> In addition, the result regarding to the alignment was added as Table 3.

- 165: how were “near complete union” and “partial union” defined?

-> Thank you for your comment. Authors have included each definition of union in the Methods section according to your recommendation as follows:

 “Serial ankle plain radiographs and a CT scan, which was obtained at 4 months after operation, were assessed to check the alignment and bony union status. Bony union was confirmed radiographically by observing the presence of trabecular lines between the tibia and the talus at the point of contact, and the disappearance of the radiolucent line [19]. Partial union was defined as partial osseous bridging formation (<70%) in the tibiotalar joint but with a significant radiolucent gap. Near-complete union was defined as a demonstrable osseous bridging in the sagittal and coronal views but with no complete absence of a radiolucent line.”

- 185: how was “talonavicular joint or subtalar joint arthritis” defined/assessed?

-> Thank you for your comment. Authors have included the information in the Materials and methods section according to your recommendation as follows:

 “Adjacent joint arthritis was defined as the appearance of joint space narrowing or osteophyte formation on standing foot and ankle radiographs.”

Discussion:

- Generally well written except for quite a number of language errors that should be edited by a native speaker.

-> Thank you for your comment. The revised manuscript was checked and polished by a native English speaker.

- Very nice figures!

-> Thank you for your comment.

Reviewer #2: 

- I would recommend to look through the manuscript regarding some grammatical errors. There are minor, but manageable.

-> Thank you for your comment. The revised manuscript was checked and polished by a native English speaker.

- Despite this, I would add the proper indications for this technique. Of course, contraindications and what could be expected from transfibular fusions (i.e. is there any chance for a conversion back into TAR?).

-> Thank you for your comment. Authors have revised the manuscript in Discussion according to your recommendation as follows: 

“On the other hand, some possible concerns remain. First, although our data showed good union rate between the tibia and talar interface, union problems at other interfaces such as the tibiofibular and talofibular space may arise. Thus, further studies are needed. Second, metal failure may occur. Lee et al. also reported a case of instrument breakage [22]. In such situations, removing the broken screw and refixing the screw may be needed, or revision surgery with another plate is an option. With regard to the revision, no single useful instrument has been established. As in our technique, the remnant posterior half fibula may serve as a reliable support. The shearing force may be limited by the remaining posterior fibula. Thordarson et al. pointed out the importance of fibular strut grafts, which provide additional stability and resistance to rotational forces [27]. Third, even if our modified technique showed no newly developed talonavicular joint or subtalar joint arthritis, this may occur in long-term follow-up [30-32], and subsequent adjacent joint arthrodesis may be needed. Lastly, use of our technique could lead to takedown of ankle fusion and conversion to total ankle replacement. Greisberg et al. reported the poor result of conversion to total ankle replacement when the lateral malleolus was resected at the time of previous fusion [33]. Our technique of preserving the posterior fibula may reduce the risk of complications when conversion to total ankle replacement is needed.

Therefore, we suggest that this modified technique is not only for general indications requiring the conventional transfibular approach, where poor soft tissue quality at the anterior ankle may be applicable, but also for patients who want to maintain the lateral malleolar contour. However, the contraindications of this technique might include soft tissue defects in the lateral malleolar area or cases of severe distal fibular deformity, for which anterior half fibulectomy and onlay graft fixation may not be feasible.”

- The technique, however, is of great interest. In our institution we have used the same technique for years due to same concerns as reported by the authors. I have to congratulate the authors to report their results and detailed technique.

-> Thank you for your comment.

---

## [Editor Report · Decision Letter 1]

9 Oct 2020

A modified transfibular technique of ankle arthrodesis using partial fibular resection and onlay bone graft

PONE-D-20-16145R1

Dear Dr. Lee,

We’re pleased to inform you that your manuscript has been judged scientifically suitable for publication and will be formally accepted for publication once it meets all outstanding technical requirements.

Kind regards,

Hans-Peter Simmen, M.D., Professor of Surgery

Academic Editor

PLOS ONE
---

## [Editor Report · Acceptance letter]

13 Oct 2020

PONE-D-20-16145R1 

A modified transfibular technique of ankle arthrodesis using partial fibular resection and onlay bone graft 

Dear Dr. Lee:

I'm pleased to inform you that your manuscript has been deemed suitable for publication in PLOS ONE. Congratulations! Your manuscript is now with our production department. 

Kind regards, 

on behalf of

Dr. Hans-Peter Simmen 

Academic Editor

PLOS ONE